# Thresholds of Endoglin Expression in Endothelial Cells Explains Vascular Etiology in Hereditary Hemorrhagic Telangiectasia Type 1

**DOI:** 10.3390/ijms22168948

**Published:** 2021-08-19

**Authors:** Georgios Galaris, Kévin Montagne, Jérémy H. Thalgott, Geoffroy J. P. E. Goujon, Sander van den Driesche, Sabrina Martin, Hans-Jurgen J. Mager, Christine L. Mummery, Ton J. Rabelink, Franck Lebrin

**Affiliations:** 1Einthoven Laboratory for Experimental Vascular Medicine, Department of Internal Medicine (Nephrology), Leiden University Medical Center, 2333 ZA Leiden, The Netherlands; G.Galaris@lumc.nl (G.G.); J.H.Thalgott@lumc.nl (J.H.T.); G.J.P.E.Goujon@lumc.nl (G.J.P.E.G.); A.J.Rabelink@lumc.nl (T.J.R.); 2Department of Mechanical Engineering, Graduate School of Engineering, The University of Tokyo, Tokyo 113-8654, Japan; montagne@m.u-tokyo.ac.jp; 3Centre for Discovery Brain Sciences, University of Edinburgh, Edinburgh EH8 9XD, UK; S.vandenDriesche@ed.ac.uk; 4ZJU-UoE Institute, Zhejiang University, Haining 314400, China; 5Center for Interdisciplinary Research in Biology (CIRB), UMR7241/U1050, 75005 Paris, France; Sabrina.martin@college-de-france.fr; 6Department of Pulmonology, St Antonius Hospital, 3435 CM Nieuwegein, The Netherlands; j.mager@antoniusziekenhuis.nl; 7Department of Anatomy and Embryology, Leiden University Medical Center, 2333 ZA Leiden, The Netherlands; C.L.Mummery@lumc.nl; 8Physics for Medicine, ESPCI, INSERM U1273, CNRS, 75012 Paris, France; 9MEMOLIFE Laboratory of Excellence and PSL Research University, 75005 Paris, France

**Keywords:** endoglin, hereditary hemorrhagic telangiectasia, cell signaling, endothelial cells

## Abstract

Hereditary Hemorrhagic Telangiectasia type 1 (HHT1) is an autosomal dominant inherited disease characterized by arteriovenous malformations and hemorrhage. HHT1 is caused by mutations in *ENDOGLIN*, which encodes an ancillary receptor for Transforming Growth Factor-β/Bone Morphogenetic Protein-9 expressed in all vascular endothelial cells. Haploinsufficiency is widely accepted as the underlying mechanism for HHT1. However, it remains intriguing that only some, but not all, vascular beds are affected, as these causal gene mutations are present in vasculature throughout the body. Here, we have examined the endoglin expression levels in the blood vessels of multiple organs in mice and in humans. We found a positive correlation between low basal levels of endoglin and the general prevalence of clinical manifestations in selected organs. Endoglin was found to be particularly low in the skin, the earliest site of vascular lesions in HHT1, and even undetectable in the arteries and capillaries of heterozygous endoglin mice. Endoglin levels did not appear to be associated with organ-specific vascular functions. Instead, our data revealed a critical endoglin threshold compatible with the haploinsufficiency model, below which endothelial cells independent of their tissue of origin exhibited abnormal responses to Vascular Endothelial Growth Factor. Our results support the development of drugs promoting endoglin expression as potentially protective.

## 1. Introduction

Hereditary Hemorrhagic Telangiectasia (HHT) is an inherited autosomal dominant vascular disorder with a prevalence of 1 in 5000 to 1 in 8000, theoretically affecting between 950,000 to 1,500,000 individuals worldwide [1]. HHT is, therefore, now relatively common, with its previous designation as rare disease probably resulting from underdiagnosis. HHT patients develop Arteriovenous Malformations (AVMs) that form when arterioles and veins connect directly without intervening blood capillaries. These vascular lesions can cause life-threatening complications such as hypoxemia, stroke, brain abscess, heart failure and fatal hemorrhage [1,2]. Many patients with HHT also suffer spontaneous, recurrent and severe nose and gastrointestinal bleeding, requiring regular iron and blood transfusions. The bleeding occurs because small AVMs (also called telangiectases) found in the skin, nasal septum, oral mucosa and in the gastrointestinal tract are particularly prone to hemorrhage [1,2].

Most cases of HHT are caused by mutations in *ENG* (endoglin) [3] or *ACVRL1* (activin receptor like kinase 1, ALK1) [4] leading to HHT type 1 (HHT1) or HHT type 2 (HHT2), respectively. The clinical phenotypes are similar but HHT2 has lower penetrance and later onset and the frequency of pulmonary and cerebral AVMs is higher in HHT1 families [5,6].

Endoglin is thought to modulate multiple signaling pathways in endothelial cells, although how these signaling pathways are affected in the context of HHT and whether there are differences between tissues remains unknown. Endoglin is an ancillary Transforming Growth Factor-β (TGF-β)/Bone Morphogenetic Protein 9/10 (BMP9/10) receptor that has no signaling kinase domain itself but can signal through ALK1 to modulate endothelial cell proliferation and migration [7,8,9,10,11,12,13,14]. Endoglin controls vascular tone by modulating eNOS expression and protein stability [15,16]. Accumulating evidence suggests a crosstalk between Vascular Endothelial Growth Factor (VEGF) and Endoglin. Patients with HHT have increased serum or plasma VEGF compared to healthy donors [17,18,19,20]. Genetic depletion of endothelial *Eng* in adult mice has been associated with high VEGF signaling activity [21] while endoglin has been reported to directly modulate VEGF Receptor 2 (VEGFR2) recycling and expression [22,23]. Blocking BMP9/10 ligand binding or genetic deletion of endothelial *Acvrl1* has been shown to induce VEGF- phosphatidyl inositol 3-kinase/protein kinase B (PI3Kinase/AKT) activity [24]. HHT2 patients and *Acvrl1^+/−^* mice that develop some features of HHT2 have a marked reduction of VEGF Receptor 1 (VEGFR1) expression that has been correlated with abnormal VEGFR2 activity and blood vessel destabilization [25].

All gene mutations, whether deletion, insertion or missense mutations, as well as splice site changes, give rise to null alleles; the mutated genes are, thus, functionally inconsequential. Haploinsufficiency is therefore widely accepted as the underlying mechanism for HHT1, the remaining wild-type allele being unable to produce sufficient protein at the cell surface to sustain endothelial cell functions [1,2]. However, the haploinsufficiency model does not explain the high inter and intra familial variation in clinical manifestations, the age of onset of the disease and why only selected vascular beds are affected, as the mutated genes are present in vasculature throughout the body. The most recent studies suggest that disease mutations are deleterious predominantly during some forms of angiogenesis [21,26,27,28] and that inflammation, infection, mechanical stress or trauma might account for the development of vascular lesions locally [25,27,29,30,31,32]. An additional somatic mutation in the remaining wild-type allele has been proposed to explain the clinical observation that vascular malformations in HHT occur focally [33]. Alternatively, inflammation has been suggested to generate a local and transient endoglin-null phenotype [26]. Several modifier genes have also been associated with the presence of pulmonary AVMs in HHT. These include *PTPN14* [34], *ADAM17* [35] and more recently some genetic variations in the functional wild-type allele of *ENG* [36].

We have quantified endoglin expression levels in blood vessels of multiple organs in mice and in humans. This revealed a positive correlation between low basal levels of endoglin and the vascular beds where lesions typically appear. We found that there appears to be a critical threshold of endoglin below which endothelial cells respond inadequately to extracellular signals: tissues with basal high levels of endoglin are protected from HHT1 mutations since the remaining wild-type *Eng* allele is still able to produce enough protein to sustain endothelial cell function. In tissues where basal levels are low, reduction due to haploinsufficiency brings levels below a critical threshold. Our data support a model in which drugs stimulating endoglin expression could prevent HHT1.

## 2. Results

### 2.1. Endoglin mRNA Expression Levels in Various Organs

The accurate quantification of endoglin levels in multiple organs is not straightforward and requires tissue-specific multistep endothelial cell isolation protocols that could all alter endoglin expression levels. As endoglin expression is mainly restricted to the endothelial cells, we have therefore chosen an alternative method consisting of quantifying mRNA endoglin expression levels in whole tissues and have used GeNorm to identify a pool of stable endothelial reference genes for normalization. The endoglin levels were measured in multiple organs collected from 9-week-old C57BL/6J and 129/ola wild-type mice. The rationale for this is that the 129/ola genetic background is more prone to develop vascular anomalies similar to those found in HHT patients. As modifier genes implicated in the regulation of TGF-β1 expression levels have been proposed to explain the susceptibility of the 129/Ola background to HHT [37] we hypothesized that endoglin levels may vary between these two genetic backgrounds, as TGF-β1 regulates endoglin expression [38]. For the selection and evaluation of stable reference genes to accurately analyze endoglin mRNA expression, we first included five commonly used endothelial cell markers (*Flk1*, *Pecam1*, *Tie2*, *Icam2* and *Tie1*). To compare the transcriptional profiles of the five candidate-reference genes across mouse tissues, the threshold (Ct) values were plotted directly (Figure 1A). The candidate reference genes exhibited variable patterns with quantitative rather than qualitative differences between tissues. The mean expression range of the 5 candidate reference genes was calculated from raw Ct values and ranged from 17.97 cycles for *Pecam1* in lung to 26.44 cycles for *Flk1* in intestine. The maximum expression range across the tissues was 7 cycles for *Flk1* and the minimum was 5.1 cycles for *Icam2*. The five candidate reference genes for normalization were ranked according to their expression stability M values using the GeNorm software. The *M* value is defined as the average pairwise variation of a certain gene with all other tested candidate-reference genes. Consequently, genes with low *M* values have a low variation and a stable expression, while genes with high *M* values have a high variation and a less stable expression. *M* values of *Tie1*, *Tie2* and *Icam2* were the lowest (0.476, 0.525 and 0.575, respectively) while *Pecam1* and *Flk1* were the highest (0.675 and 0.927, respectively) indicating that *Tie1*, *Tie2* and *Icam2* had the most stable expression across tissues (Figure 1B).

Next, we evaluated the optimal number of candidate reference genes that were needed for accurate normalization. Pairwise variations V_n_/V_n+1_ between two sequential normalization factors (NF) were calculated to determine the effect of adding the next candidate reference gene in normalization. A large variation implies that the added gene has a significant effect and should preferably be excluded for the calculation of a reliable NF [39]. Based on the pairwise variation analysis, we concluded that three candidate-reference genes were sufficient under our conditions, although we did not reach the threshold of 0.15 (Figure 1C).

Endoglin levels were then measured and three endothelial reference genes. *Tie1, Tie2* and *Icam2*, determined by GeNorm were used to normalize its expression across all sample sets. The endothelial cells showed striking heterogeneity in the levels of endoglin that depended on the organ in which they were present. The endoglin levels were low in skin, lung, intestine and brain, organs described as being susceptible to the development of vascular malformations in patients with HHT1, compared to heart, kidney and liver (Figure 1D,E). No significant differences were found between the two genetic backgrounds, indicating that variations in TGF-β1 expression do not influence basal levels of endoglin expression in tissues, at least under physiological conditions (Figure 1D).

### 2.2. Organs with Basal Low Levels of Endoglin Associate with High Risk to Develop HHT

We next used confocal fluorescence microscopy to quantify endoglin expression levels accurately in the capillaries of various organs from 9-week-old C57Bl/6J wild-type mice (Figure 2). All samples were prepared in parallel to increase precision in the quantifying fluorescence signals. The sample preparations included tissue isolation, fixation, permeabilization, labeling and mounting. Pecam1 staining was used to label vascular structures and showed relatively homogeneous expression levels across tissues (Figure 2A). The images were acquired using Leica SP5 confocal microscope. We used low laser power and fast scan speeds to minimize photobleaching and saturation of fluorophores. The signal of the images was relatively high compared to the noise level (Figure 2A,B). A range of photomultiplier tube (PMT) voltages were tested to delineate the region of linearity between the fluorescence intensity and the relative expression of endoglin. Skin and liver sections were compared as tissues expressing the lowest and highest levels of endoglin, respectively, and the PMT voltage was selected so that all samples were within the linear range for quantification [40,41] (Figure 2C). As expected, endoglin protein levels were found to be particularly low in skin, lung, intestine and brain, confirming the real-time RT-PCR data (Figure 2D).

### 2.3. Endoglin Levels Become Almost Undetectable in Skin Capillaries and Arteries of Eng^+/−^ Mice

We next examined various organs in endoglin heterozygous mice (*Eng^+/−^)* that are the closest genetic model of HHT1 in terms of genotype. These mice exhibited an intermediate phenotype with a widespread abnormality of the vascular walls and development of age-dependent vascular lesions similar to those seen in HHT patients [42,43,44]. As expected, endoglin levels were reduced by half in all analyzed organs (Figure 3A).

To further characterize endoglin expression along the arteriovenous axis, we stained skin of the dorsal ear of 8-week-old wild-type and *Eng^+/−^* mice for endoglin or Platelet Endothelial Cell Adhesion Molecular 1 (PECAM1) and for α-smooth muscle actin (α-SMA) that labels vascular smooth muscle cells (VSMCs). The overall vascular pattern was similar between wild-type and *Eng^+/−^* mice with arteries and veins covered by VSMCs that follow parallel routes and with the presence of a dense capillary network (Figure 3B). While PECAM1 was uniformly expressed in all endothelial cells, the expression of endoglin varied along the arteriovenous axis. Endoglin was predominantly found in veins and venules while its expression was particularly low in capillaries and in arteries (Figure 3B) confirming that heterogeneity in endoglin expression also exists within a particular tissue. Interestingly, endoglin expression became almost undetectable in capillaries and arteries in *Eng^+/−^* mice (Figure 3B) supporting the notion that local loss of endoglin is necessary but alone is not sufficient for AVM formation [45].

### 2.4. Levels of Endoglin Are Linked to Tissue Developing HHT1 in Humans

To extend our data to humans, sections of various organs isolated from one 57-year-old HHT1 donor and stained with PECAM1 and endoglin revealed that endoglin levels were lower in blood vessels of all principal tissues affected by HHT compared with kidney or heart (Figure 4A,B).

### 2.5. Signaling Pathways in ECs Are Dependent on Endoglin Thresholds

We next tested the hypothesis that endothelial cells might be sensitive to a critical endoglin threshold below which they will respond inadequately to their environment, independently of their tissue of origin. For this, primary endothelial cells were isolated from the lungs and liver of *Eng^flox/flox^* pups at postnatal day 7 (P7) using collagenase I-based enzymatic digestion followed by cell sorting with PECAM1-coated microbeads [46] (Figure 5A). CD45-coated microbeads were used to deplete the CD45^+^/PECAM1^+^ immune cell population (Figure 5A). The isolated CD45^-^/PECAM1^+^ cell populations resembled that of endothelial cells and could be cultured for at least three passages. PECAM1 and VE-cadherin staining confirmed the endothelial cell identity (Figure 5B). Heterogeneity in endoglin levels was observed among tissue-specific endothelial cells reflecting the presence of arterial, capillary and venous populations (Figure 5B,D). Endoglin expression was higher in liver than in lung endothelial cells, suggesting that endoglin levels throughout the body are not only regulated by the tissue environment, but also epigenetically (Figure 5B,D,E). Our results also indicated that variation in endoglin levels between tissues could be maintained in vitro for several passages as also shown for other endothelial cell markers [47,48]. Taking advantage of the Cre-Lox system, primary endothelial cell cultures were infected by recombinant adenoviruses encoding the Cre recombinase to excise the *Eng* gene. Adenoviral infection was highly efficient and had no effect on cell viability [49].

Almost all cells underwent gene recombination within the first 10 h and as endoglin is a relatively stable protein with a half-life estimated as around 17 h [50], homogeneous endothelial cell cultures were obtained within the first 60 h after infection with endoglin levels ranging from normal and half reduced to complete loss as shown by staining and Western blot analyses (Figure 5C–E). To investigate how lung and liver endothelial cells responded to endoglin thresholds, we next investigated the activation of the Akt and Smad1 signaling pathways. Lung and liver endothelial cells were treated with VEGF, TGF-β1 and BMP9 for 30 min at 6, 20 or 60 h after infection that corresponded to normal, half reduction and loss of endoglin expression (Figure 5E). As expected, liver and lung endothelial cells with endoglin levels decreased by >90% showed an increase in Akt phosphorylation and a decrease in Smad1 phosphorylation while the total levels of Akt and Smad1 were unchanged (Figure 5E). However, since liver endothelial cells had relatively high basal expression of endoglin, reduction by half did not affect Akt or Smad1 activity in response to VEGF and TGF-β1/BMP9, respectively. In contrast, lung endothelial cells that had low basal levels of endoglin were highly sensitive to endoglin thresholds. Reduction by half induced an increase in Akt phosphorylation compared to lung endothelial cells expressing normal levels of endoglin (Figure 5E). Smad1 phosphorylation was unchanged. We analyzed earlier time points after infection to determine the minimal threshold below which lung endothelial cells become sensitive to VEGF and confirmed that a reduction of endoglin expression by half is indeed required for endothelial cells to respond inadequately to VEGF stimulation, as shown by an increase in Akt and p42/p44 Mitogen Activated Protein Kinase (MAPK) phosphorylation (Figure 5E).

## 3. Discussion

Although HHT is caused by germline aberrations in genes that are expressed in all blood vessels, vascular malformations only develop in a remarkably limited number of tissues across the body. Unravelling mechanisms that govern tissue-specific manifestations is therefore critical to understand HHT etiology and direct efforts to develop treatments. Here, we have quantified by real-time RT-PCR and immunofluorescent staining the expression levels of endoglin in the blood vessels of multiple tissues in mice and in humans and revealed an association between tissues that express low levels of endoglin in vivo and the general prevalence of vascular malformations developing in specific organs in HHT1. We have also shown that endoglin levels are not associated with organ-specific vascular functions and that endoglin is apparently not functionally more important where it is highly expressed. Instead, we have demonstrated that there is a critical threshold of endoglin below which endothelial cells respond inadequately to their environment and that tissues with low basal levels of endoglin are highly sensitive to HHT1 mutations. Importantly, we show that VEGF-Akt signaling pathway is highly sensitive to reduced endoglin levels confirming that targeting VEGF has potential value in treating patients with HHT.

Endoglin expression levels were found to be exceptionally low in skin, a tissue with notably high prevalence of telangiectases even at young age [51,52]. Lung, brain and intestine also expressed low levels of endoglin compared with heart, liver and kidney, implying by extrapolation that these tissues may be more susceptible to developing HHT1. Several studies have examined the presence and distribution of vascular malformations in large groups of HHT patients to evaluate phenotypic differences between *ENG* and *ACVRLI* mutation carriers. Consistent with our data, whilst mucocutaneous telangiectases and GI bleeding were found in both groups, HHT1 was more frequently associated with congenital cerebral and pulmonary AVMs whereas HHT2 predominantly involved the liver [6,53,54,55,56].

We have found that endoglin is strongly expressed in veins compared to arteries and blood capillaries [22,26] and noted that its expression as almost undetectable in arteries and blood capillaries of skin isolated from endoglin heterozygous mice. This suggests arteries and/or blood capillaries as the origin of the development of vascular anomalies in HHT1. Several lines of evidence support this idea. Analysis of mosaic *Eng* Loss of Function retinal vasculature in mice has demonstrated that AVMs predominantly arise from arterioles [22]. The expression levels of some arterial markers such as Ephnb2 are reduced in the absence of endoglin [22]. However, a recent study has convincingly demonstrated that the loss of arterial endoglin was not involved in driving the formation of AVMs in mice, implying that capillaries might be the blood vessels playing a critical role in this pathological process [57]. Our findings support the idea that an almost complete loss of endoglin is a prerequisite for the development of vascular lesions as previously suggested [26,45], but also implies that a local transient release of the receptor from the cell surface during inflammation or a second local mutation in the wild-type allele might not be necessary for the development of vascular malformations, at least in tissues such as the skin [26,33]. We propose instead that heterogenous mutations in the *Eng* gene are sufficient to drive endothelial cell dysfunction, notably a high sensitivity to VEGF in organs with low basal levels of endoglin, such as the skin, and that these organs might be more susceptible to develop vascular malformations in presence of angiogenic stimuli. In agreement, angiogenic or inflammatory triggers have been reported to promote the formation of vascular lesions in *Eng* heterozygous mice, particularly in organs where we found endoglin was low, as in the brain or the intestine [29,58,59].

Our results might also explain the clinical variability within and between families. Since most causative *ENG* mutations are loss of function, genetic variants within the mutated gene are likely to be functionally inconsequential. However, genetic variation within the wild-type allele inherited from the healthy parent might result in substantial variations in endoglin levels in HHT patients. As consequence, individuals with the same mutation even within the same family may exhibit major differences in clinical features of the disease based on which tissues endoglin decreases below the threshold. A recent study demonstrating a genetic association between particular Single Nucleotide Polymorphisms spanning the *ENG* gene and the prevalence of pulmonary AVMs supports this idea. Interestingly, they also found that the AVM-protective allele was associated with higher expression of endoglin [36].

To explain why only selected tissues are affected in HHT, we propose a model in which endothelial cells, independent of their tissue of origin, exhibit abnormal functions only if endoglin levels fall below a critical threshold. Our study supported indications that heterogeneity in endoglin levels might depend on environmental cues such as cytokines, growth factors, metabolites, biophysical signals and direct cell–cell contact with parenchyma cells [47,48]. Organ-specific expression of endoglin might also be under epigenetic control as we revealed that cultured lung and liver endothelial cells showed tissue differences in endoglin levels and as for other endothelial cell markers, this was maintained even after passage in culture [60]. Our model implies that tissues expressing low basal levels of endoglin, such as the lungs, are more sensitive to HHT1 mutations whereas endothelial cells expressing high levels of endoglin such as the liver are protected from abnormalities arising from HHT1 mutations as the remaining wild-type allele produces sufficient protein to support normal endothelial cell function (Figure 6).

We also found that lung endothelial cells endoglin expression reduced by half became highly sensitive to VEGFA stimulation augmenting Akt and MAP kinase activities whereas Smad1 phosphorylation induced by TGF-β/BMP9 was unchanged. In contrast, loss of endoglin by ≥90% in both lung and liver endothelial cells decreased Smad1 phosphorylation induced by TGF-β/BMP9 and increased Akt activity induced by VEGF. Our findings confirm the critical role of VEGF signaling in the etiology of HHT1 [21,22]. Moreover, our study suggests that the difference between VEGF and TGF-β/BMP9 signaling pathways to endoglin thresholds may depend on high and low affinity interactions between endoglin and TGF-β/BMP receptors versus VEGFR2. We hypothesize that endoglin may have a lower affinity for VEGFR2, implying that this signaling pathway will be affected first by reduced endoglin levels (Figure 6). Our results also imply that mutations in endoglin lead to increased levels of Akt only in selected vascular beds that in turn might be more prone to respond inadequately to angiogenic triggers favoring the formation of vascular anomalies.

In conclusion we provide novel insights into the mechanisms underlying the development of AVMs in specific organs and confirm the key role of VEGFR2 signaling in the etiology of HHT1. Moreover, our results support the development of drugs stimulating the expression of endoglin as a protective agent against HHT1 mutations.

## 4. Materials and Methods

### 4.1. Human Material

Biopsies from one HHT1 donor were obtained from the Department of Pulmonary Disease of the St Antonius Hospital following informed consent. All procedures were reviewed and approved by the medical ethics committee of the St Antonius Hospital, Nieuwegein, The Netherlands. The investigation conforms to the principles outlined in the Declaration of Helsinki. Molecular diagnosis was based on identification of the *ENG* mutation. Biopsies were collected in physiological salt, fixed overnight in 0.2% paraformaldehyde in 0.1 M phosphate buffer with 0.12 CaCl_2_ and 4% sucrose and then processed as described [44]. We used 7 µm sections for staining.

### 4.2. Mice

Mice were maintained under standard specific pathogen-free conditions and all animal procedures performed were reviewed and approved by the Institutional Committees for Animal Welfare of Ile de France and Leiden University Medical Center (project numbers 2014_19_2041.01 and AVD1160020171628, respectively) and followed the French and Dutch government guidelines and directive 2010/63/EU of the European Parliament. C57Bl/6J and 129/ola males were used at eight to nine weeks of age. *Eng^flox/flox^* mice [61] were kindly provided by Helen M. Arthur and pups were used during the first postnatal week of life.

### 4.3. Mouse Endothelial Cell Isolation and Culture

Lungs and liver from *Eng^flox/flox^* pups were surgically removed, rinsed in ice-cold DMEM and endothelial cells were isolated as described [46]. Briefly, tissues were minced by the use of scissors, digested in DMEM-3 mg.ml^−1^ Collagenase A (10103586001, Roche, Basel, Switzerland) for 15 min at 37 °C and then filtered through a 70-µm strainer. The cell suspension was centrifuged at 200× *g* for 5 min and CD45^+^ cells were removed using Dynabeads sheep anti-Rat IgG (11035, Invitrogen, Waltham, MA, USA) coated with rat anti-mouse CD45 antibody (550539, BD Pharmigen, Franklin Lake, NJ, USA). Endothelial cells were sorted using Dynabeads sheep anti-Rat IgG (11035, Invitrogen, Waltham, MA, USA) coated with rat anti-mouse PECAM1 antibody (550274, BD Pharmigen, Franklin Lake, NJ, USA) according to the manufacturer’s instructions. After washing 5 times with DMEM-0.1% BSA, cells were seeded in 6-well plates. Lung and liver ECs were maintained for two-three passages in Endothelial Cell Growth Medium 2 (C-22011, PromoCell, Heidelberg, Germany) complemented with Fetal Calf Serum, Human Epidermal Growth Factors, Basic Fibroblast Growth Factor, Insulin Like Growth Factor, Human Vascular Endothelial Growth Factor-165, Ascorbic acid, Heparin and Hydrocortisone (C-39211, SupplementPack EC GM2, PromoCell, Heidelberg, Germany).

### 4.4. Adenoviral Infection

Lung or liver endothelial cells were infected with an adenovirus expressing the Cre-Recombinase (SL100707, SignaGen Laboratories, Rockville, MD, USA) using a multiplicity of infection (m.o.i) of 500 for 6 h, washed with PBS and then cultured in Endothelial Cell Growth Medium 2 for 12 to 60 h before use.

### 4.5. Immunofluorescence Staining

Cultured mouse endothelial cells were stained as described [25]. Briefly, cells were fixed with cold-ethanol for 10 min and then blocked in PBS with 2% BSA (A9418, Sigma-Aldrich, Saint Louis, MO, USA) for 1 h at room temperature. The following primary antibodies were used in blocking solution: rat anti-mouse endoglin (120401, BioLegends, San Diego, CA, USA), rabbit anti-mouse VE-Cadherin (sc-28644, Santa Cruz), goat anti-mouse Pecam1 (sc-1506, Santa Cruz) or mouse anti-human endoglin antibody (clone 266) (555690, BD Pharmingen, Franklin Lake, NJ, USA). After washing, coverslips were incubated with Alexa Fluor^®^ 555 donkey anti-rabbit (A-32794, Invitrogen, Waltham, MA, USA), with Alexa Fluor^®^ 488 donkey anti-rat (A-21208, Invitrogen, Waltham, MA, USA), Alexa Fluor^®^ 647 donkey anti-goat (A-21447, Invitrogen, Waltham, MA, USA) or Alexa Fluor^®^ 555 goat anti-mouse (A32727, Invitrogen, Waltham, MA, USA). Nuclei were counterstained with Dapi (D1306, Invitrogen, Waltham, MA, USA) before mounting.

Mouse or human tissue sections were processed as described [25]. Briefly, frozen sections were fixed in cold acetone for 10 min, washed with PBS, blocked with and incubated overnight at 4 °C with primary antibodies in PBS with 0.2% Bovine Serum Albumin (BSA) (A9418, Sigma-Aldrich, Saint louis, MO, USA).

Cryosections of mouse tissues were labelled with rat anti-mouse endoglin (120401, BioLegends) or rat anti-mouse PECAM1 (Clone 13.3) (550274, BD Pharmigen, Franklin Lake, NJ, USA). After washing, sections were incubated for 1 h at room temperature with the secondary antibody Alexa Fluor^®^ 555 goat anti-mouse (A32727, Invitrogen, Waltham, MA, USA) or Alexa Fluor^®^ 555 donkey anti-rat (A-21434, Invitrogen, Waltham, MA, USA) diluted in PBS with 0.2% BSA (A9418, Sigma-Aldrich, Saint louis, MO, USA) and were then mounted in DAKO mounted medium (S302380-2, DAKO, Copenhagen, Denmark). We captured images with a confocal laser-scanning microscope SP5 (Leica, Wetzlar, Germany).

Cryosections of human HHT1 tissues were labelled with mouse anti-human endoglin antibody (clone 266) (555690, BD Pharmingen, Franklin Lake, NJ, USA) or goat anti-human PECAM1 antibody (clone M-20, Santa Cruz Biotechnology, Santa Cruz, CA, USA) and then incubated with Alexa Fluor^®^ 555 goat anti-mouse (A32727, Invitrogen, Waltham, MA, USA) or Alexa Fluor^®^ 555 rabbit anti-goat (A21431, Invitrogen, Waltham, MA, USA).

For whole mount skin staining, ears were separated with forceps into dorsal and ventral leaflets. Dorsal halves were fixed in 4% PFA overnight at 4 °C and subsequently permeabilized by incubation in PBS-triton X100 0.2% (Sigma-Aldrich, Saint louis, MO, USA) for 2 h at room temperature and then blocked by incubation in blocking reagent (Roche-11096176001) for 2 h at room temperature. The ear microvasculature was labelled with rat anti-mouse Pecam1 (Clone 13.3) (550274, BD Pharmigen, Franklin Lake, NJ, USA) or with rat anti-mouse endoglin (clone 2Q1707) (sc-71042, Santa Cruz Biotechnology, Santa Cruz, CA, USA) and then incubated with Alexa Fluor^®^ 555 donkey anti-rat (A-21434, Invitrogen, Waltham, MA, USA) and with monoclonal anti-α-smooth muscle actin conjugated with FITC (clone 1A4) (F3777, Sigma-Aldrich, Saint louis, MO, USA).

### 4.6. Image Quantification

Digital images of fluorescent stained sections from various tissues were acquired with a Leica SP5 confocal microscope. Image J software was used for computerized analysis of the vasculature. For each image, endoglin fluorescence intensity was calculated with the mean grey value defined as the average grey value within the selection of at least 30 blood capillaries for each organ.

The same approach was used to quantify endoglin expression in endothelial cells and at least 10 cells per time point.

### 4.7. RNA Isolation and Real Time PCR Analysis

Total RNA was isolated using TRIzol^®^ Reagent (15596-026, Invitrogen, Waltham, MA, USA) according to the manufacturer’s instructions. Samples were DNase I treated to eliminate genomic DNA and 1 µg RNA was reversed transcribed. Real-time RT-PCR was performed in a CFX-96 (BIO-RAD, Hercules, CA, USA). For mouse *Eng* mRNA levels, the stability of the candidate reference genes under a series of conditions including various organs and different genetic backgrounds was evaluated using the GeNorm software (available online: https://medgen.ugent.be/gennorm/ (accessed on 28 July 2021)). We selected conventional endothelial cell genes suitable for gene expression normalization as candidate reference genes. The genes included were *Flk1, Pecam1, Tie1, Tie2* and *Icam2*. GeNorm calculated the factor M corresponding to the stability of the reference gene, the paired variation V value of the normalized factor after introducing a new internal reference gene and the number of optimal internal reference genes [39]. Quantitative PCR was performed with an initial denaturation step of 15 min at 95 °C followed by 40 cycles of 15 s denaturation at 95 °C, 30 s annealing at 60 or 64 °C, and 30 s extension at 72 °C. A minimum of three biological replicates was carried out for all quantitative PCR reactions. Fold changes were calculated using the comparative CT method as follows:

Relative endoglin mRNA expression = 2^- (Ct endoglin—Ct reference pool) and Ct reference pool = $\sqrt[{3}]{\left(Ct\;tie1*Ct\;tie2*Ct\;ICAM2\right)\;}$. Primers and conditions are described in Table 1.

### 4.8. Western Blot and Quantification

Mouse endothelial cells were seeded in 6-well plates and allowed to grow to 90% confluence. Cells were washed with PBS and serum starved for six hours. Cells were stimulated with VEGF (25 ng.mL^−1^), TGF-β1 (1 ng·mL^−1^) and BMP9 (1 ng·mL^−1^) for 30 min, washed with cold PBS and then lysed in RIPA Buffer (50 mM Tris-HCl pH 7.4, 150 mM NaCl, 1 mM Ethylene diamine tetra-acetic acid (EDTA), 1% Triton X-100, 0.1% Sodium Dodecyl Sulfate (SDS), 0.5% deoxycholate) containing protease and phosphatase inhibitors (PPC1010, Sigma-Aldrich, Saint louis, MO, USA).

Protein concentrations were determined using a BCA protein assay kit (23225, Pierce™) and 15 μg of total protein was loaded per condition. Samples were boiled for 10 min and proteins were separated on 10% acrylamide gel and transferred onto nitrocellulose membrane before blocking with 5% BSA or 5% powdered milk/Tris-buffered saline/Tween 20 and incubated with the following primary antibodies: rabbit anti-Phospho-Akt1 (Ser473) (44-621G, Invitrogen, Waltham, MA, USA), rabbit anti-Akt1 (2938S, Cell Signaling, Danvers, MA, USA), rabbit anti-PhosphoSmad1/5 (Ser463/465) (9516S, Cell Signaling), rabbit anti-Smad1 (6944S, Cell Signaling, Danvers, MA, USA), mouse anti-Phospho-p44/42 MAPK (Thr202/Tyr204) (E10) (9106S, Cell signaling, Danvers, MA, USA), rabbit p44/42 MAPK (Erk1/2) (9102S, Cell Signaling, Danvers, MA, USA), goat anti-Endoglin (AF1097, R&D systems, Minneapolis, MN, USA) and mouse anti-β-actin (A5441, Sigma-Aldrich, Saint louis, MO, USA). Protein detection was achieved using HRP anti-rabbit IgG or HRP anti-mouse IgG (W4011 or W4021, respectively, Promega, Madison, WI, USA) or HRP anti-goat IgG (HAF017, R&D systems, Minneapolis, MN, USA) followed by scanning on BIO-RAD ChemiDoc imager. Images were captured in the linear range and ImageJ software was used for measuring protein and protein phosphorylation levels. For all blots, background was subtracted and results were normalized as indicated in the figures.

### 4.9. Statistical Analysis

We performed statistical analyses with Prism-9 software (GraphPad, San Diego, CA, USA) using 1-way ANOVA for multiple comparisons. We used for post hoc pairwise comparisons the Dunnett’s test. Results are expressed as mean ± SEM. A value of * *p* < 0.05, ** *p* < 0.01 or **** *p* < 0.01 denoted statistical significance.

## Figures and Tables

**Figure 1 ijms-22-08948-f001:**
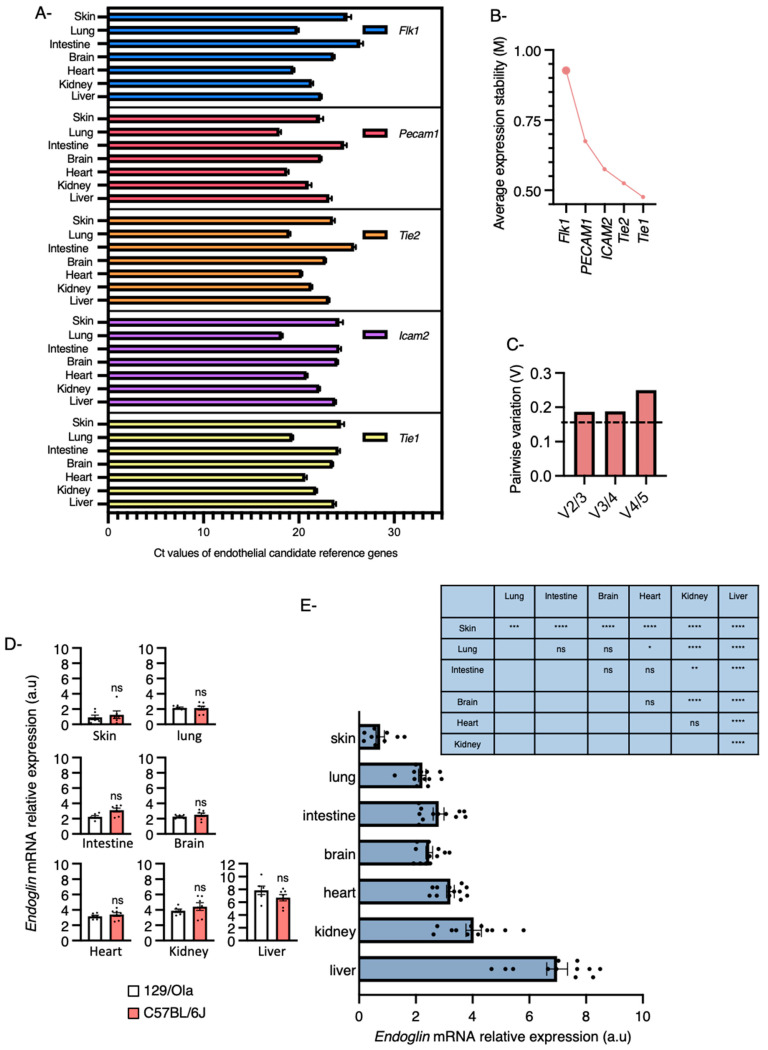
Endoglin mRNA expression levels in multiple mouse organs. (**A**) Ct values for all candidate genes in skin, lung, intestine, brain, heart, kidney and liver isolated from 9 weeks old 129/Svj (*n* = 6) and C57Bl/6J (*n* = 7) mice. (**B**) Average expression stability values (M) of the candidate reference. M is represented from the least stable (left) to the most stable (right), analyzed by GeNorm software. (**C**) Pairwise variation analysis between normalization factors to determine the optimal number of control genes for normalization. (**D**,**E**) Real time PCR for endoglin normalized to a reference pool of three endothelial reference genes (*Tie1, Tie2 and Icam2*) determined by GeNorm comparing endoglin levels in multiple tissues isolated from 9 weeks old 129/Svj (*n* = 6) and C57Bl/6J (*n* = 7) mice. (**D**) Non significant (ns) results from Mann–Whitney *U* test that compares the median of two groups. (**E**) Error bars represent SEM. * *p* < 0.05, ** *p* < 0.01, *** *p* < 0.001 and **** *p* < 0.0001 result from one-way ANOVA and Tukey’s multiple comparison tests comparing the mean of each group with the mean of the other groups. ns: Not significant.

**Figure 2 ijms-22-08948-f002:**
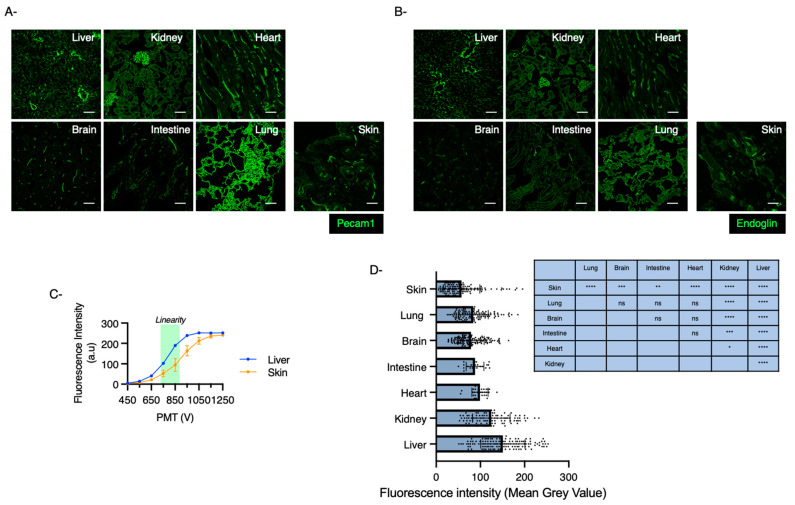
Marked heterogeneity in endoglin levels across different mouse tissues. (**A**,**B**) Sections of multiple organs isolated from 9-week-old C57Bl/6J mice stained for Pecam1 (**A**) or endoglin (**B**) (scale bar: 50 μm). (**C**) Fluorescence intensity was measured over the full range of PMT voltages for liver and the skin sections to determine the linear fluorescence intensities for each tissue. We identified the PMT gain of 850 as optimal to ensure an accurate quantification of endoglin levels across the various tissues. (**D**) Endoglin levels were quantified using ImageJ software defined as the mean grey intensity of the blood capillaries selected in each image. A minimum of 30 blood vessels was quantified per organ isolated from 3 wild-type mice. Error bars represent SEM. * *p* < 0.05, ** *p* < 0.01, *** *p* < 0.001 and **** *p* < 0.0001 result from 1-way ANOVA and Tukey’s multiple comparison tests comparing the mean of each group with the mean of the other groups. ns: Not significant.

**Figure 3 ijms-22-08948-f003:**
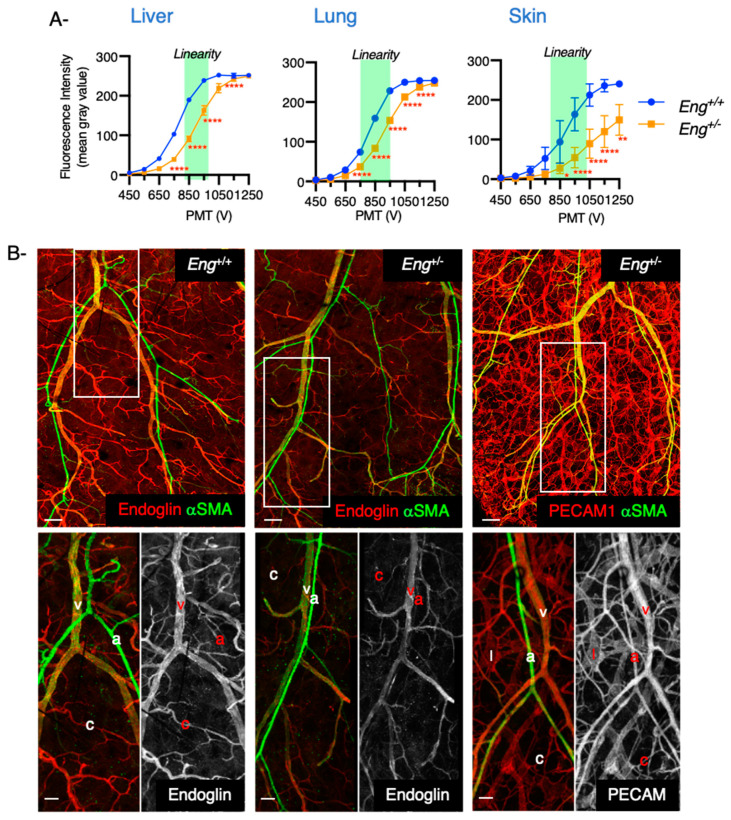
Heterogeneity in endoglin expression along the arteriovenous axis. (**A**) Fluorescence intensity was measured over the full range of PMT voltages for liver, lung and skin sections from 9-week-old C57Bl/6J wild type (*n* = 3) and *Eng^+/−^* mice (*n* = 3) and confirmed that endoglin levels were reduced by half in all organs. (**B**) Upper panels. Confocal images of whole mount ears isolated from 8-week-old wild-type and *Eng^+/−^* mice and stained for endoglin or PECAM1 and α-SMA (Scale bar: 500 μm). Lower panels are higher magnifications showing that endoglin is predominantly expressed in veins and becomes almost undetectable in arteries and capillaries in *Eng^+/−^* mice (Scale bar: 100 μm). Error bars represent SD. * *p* < 0.05, ** *p* < 0.01 and **** *p* < 0.0001 result from 2-way ANOVA and Šidàk’s multiple comparisons test.

**Figure 4 ijms-22-08948-f004:**
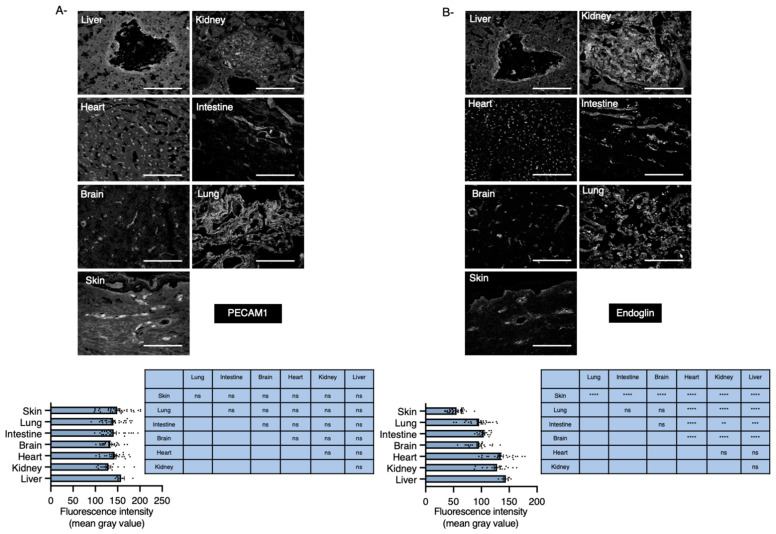
Endoglin expression levels in human tissues. (**A**,**B**) Sections of multiple tissues isolated from a 57 year-old patient with HHT1 stained for PECAM1 (**A**) or endoglin (**B**) (Scale bar: 50 μm). Pecam1 and endoglin levels were quantified using ImageJ software defined as the mean grey intensity of the blood capillaries selected in each image. Blood vessels were quantified per organ isolated from at least three independent tissue sections. Error bars represent SEM. ** *p* < 0.01, *** *p* < 0.001 and **** *p* < 0.0001 result from 1-way ANOVA and Tukey’s multiple comparison tests comparing the mean of each group with the mean of the other groups. ns: Not significant.

**Figure 5 ijms-22-08948-f005:**
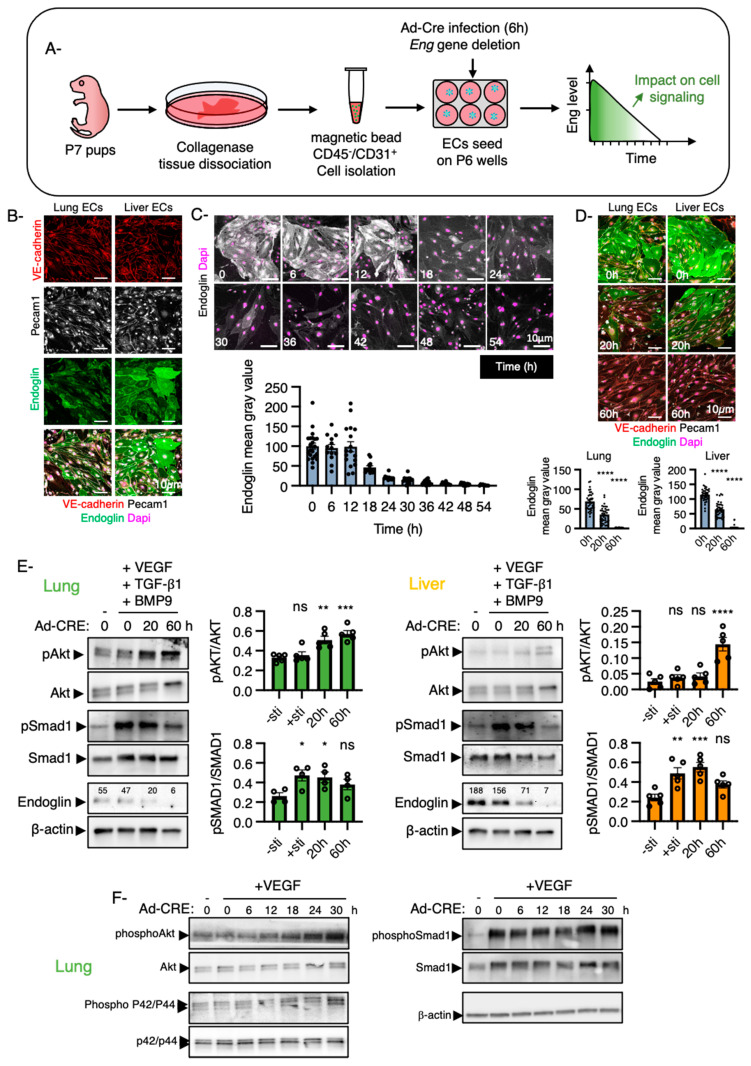
Different sensitivity of organ-specific ECs to reduced endoglin levels. (**A**) Schematic illustration of the strategy used to study how endoglin levels influence EC signaling pathways. It includes the microbeads-based protocol for the isolation of ECs [46] and the *Eng* gene deletion strategy. (**B**) Confocal images of cultured ECs isolated from lungs and liver and stained for PECAM1, VE-Cadherin and Endoglin (Scale bar: 10 μm). (**C**) Confocal images of lung ECs stained for endoglin at different time points after *Eng* gene deletion and fluorescence intensity quantification of its expression at the endothelial cell surface (a minimum of 10 cells were quantified per time point) (Scale bar: 10 μm). (**D**) Confocal images of lung and liver ECs stained for PECAM1, VE-Cadherin and Endoglin at 0, 20 and 60 h after viral infection, showing the efficacy of the *Eng* gene deletion (Scale bar: 10 μm) and fluorescence intensity quantification of endoglin expression at the endothelial cell surface (a minimum of 35 cells were quantified per time point). (**E**) ECs isolated from lungs and liver of P7 pups were exposed to VEGF (25 ng.ml^−1^), TGF-β1 (1 ng.ml^−1^) and BMP9 (1 ng.ml^−1^) for 30 min at 37 °C before lysis. Whole cell extracts were fractionated by SDS-page and blotted. The filters were incubated with Phospho-Akt, Phospho-Smad1, Akt, Smad1, endoglin and β-actin. Representative results from at least 4 independent experiments are shown. Graphs represent quantification of the Western blotting. Error bars represent SEM. * *p* < 0.05, ** *p* < 0.01, *** *p* < 0.001 and **** *p* < 0.0001 result from 1-way ANOVA and Dunnett’s post hoc tests comparing the mean of each group to the unstimulated condition. ns: Not significant. (**F**) Western blot analysis of Akt, p42/p44 MAPK and Smad1 phosphorylation at different time points after *Eng* gene deletion.

**Figure 6 ijms-22-08948-f006:**
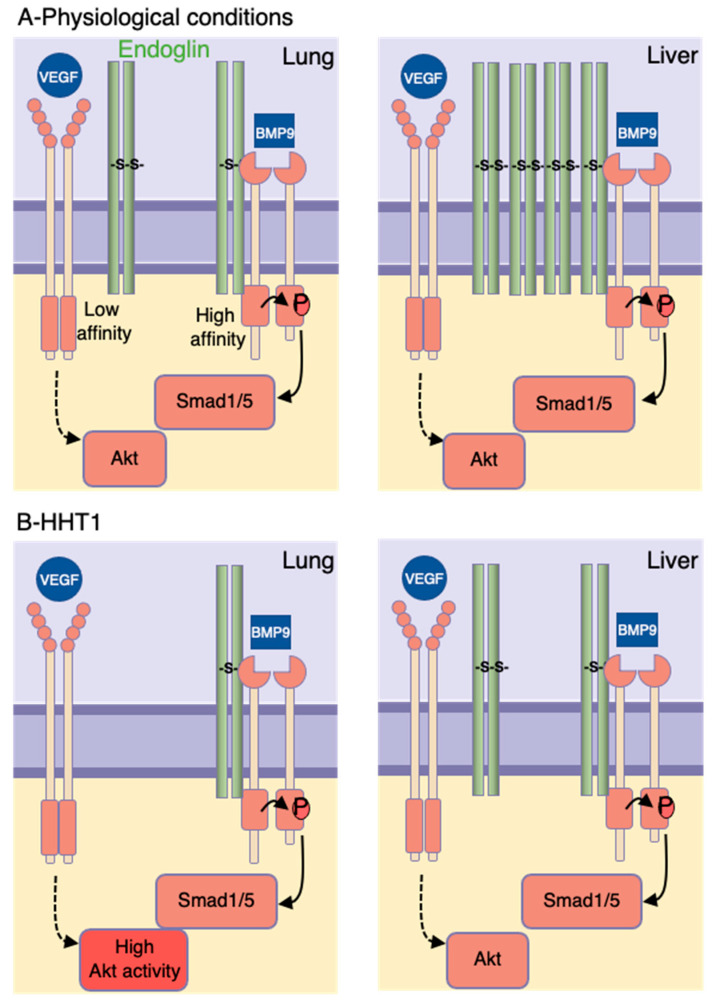
Schematic representation of how heterogeneity in endoglin levels in ECs predict the tissue-specific manifestations of HHT1.

**Table 1 ijms-22-08948-t001:** Primers and conditions.

Gene	Primer Sequence	Product Size	Annealing Temperature
*Flk1*	5′-GGCGGTGGTGACAGTATCTT-3′ 5′-GAGGCGATGAATGGTGATCT-3′	152bp	60 °C
*Tie1*	5′-CATCGAGACTTTGCAGGTGA-3′ 5′-GTTTCCATAGGGGGCGTATT-3′	132bp	60 °C
*Tie2*	5′-AAGCATGCCCATCTGGTTAC-3′ 5′-GCCTGCCTTCTTTCTCACAC-3′	138bp	60 °C
*Pecam1*	5′-GCACCCATCACTTACCACCT-3′ 5′-GCTCGTCCCCTCTTTCACA-3′	279bp	60 °C
*Icam2*	5′-CATCCTCAAGGGAAGTGGAA-3′ 5′-ACTTGAGCTGGAGGCTGGTA-3′	137bp	60 °C
*m Endoglin*	*5′-CTTCCAAGGACAGCCAAGAG-3′* *5′-GTGGTTGCCATTCAAGTGTG-3′*	221bp	60 °C

## Data Availability

Not applicable.

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
