# Peer review of "Thresholds of Endoglin Expression in Endothelial Cells Explains Vascular Etiology in Hereditary Hemorrhagic Telangiectasia Type 1"

_ijms, 2021, doi:10.3390/ijms22168948_

Round 1

Reviewer 1 Report

The experiment design is sound, the data solid, and the results clearly presented. I only found one typo on page 2, line 78. ....to produce sufficient protein at the cell surface, not as the cell surface.

Reviewer 2 Report

The work of Galaris et al. shows that there is variability in Eng expression between different tissues/organs and that this variability may explain the characteristic distribution of vascular alterations in HHT. Their conclusions are that there is a critical threshold for endoglin expression levels below which cells present alterations in the response to VEGF that would explain the development of vascular alterations.

This manuscript addresses an interesting aspect of HHT as is the high intra- and inter-patient variability in clinical manifestations. The work is interesting and easy to read. However, in my opinion, the article could be improved on a few points

Major points

  1. The first set of results is based on the analysis of Eng expression in different tissues corrected for a set of reference genes. In fact, the first part of Figure 1 refers to the search for these genes. But then those genes are not stated anywhere (or I could not find them). I assume that they are Tie1, Tie2, and ICAM2, from the results in Fig 1B, but I think it should be more explicitly stated in different places (text of the article, figure legend, and methodology).
  2. Figure 1D shows the relative expression of Eng mRNA on a scale ranging from 0 to 12, but no units are indicated. Relative with respect to what? What would be 1 in this case? Is it relative to the skin? In the skin there are samples with a value less than 1, what does it mean in that case? I think that, given the relevance for the conclusions reached, the whole process of quantification of the results of Eng expression in the different tissues should be better explained.
  3. Following on from the previous point, it is not clear whether it has been analyzed whether there is a different expression of Eng between mice with C57 and 129/Ola background. In lines 146-148 it is indicated that there are no differences between the two backgrounds, however since it is not clear with respect to what Eng expression has been referenced, it is not clear whether the data are comparable per se.
  4. The data relating the results obtained in mice to HHT in humans are somewhat poor. On the one hand, I believe that an n=1 is no more than an isolated case. On the other hand, perhaps the analysis could be completed with quantitative data as has been done in mice. Either by quantification of histologies or by analysis of gene expression by Real-Time PCR (preferably both since the work would be much more complete).
  5. In line 211, the authors state a hypothesis that will lead to one of the conclusions of the study: "high endoglin levels might not be directly associated with organ-specific vascular functions". However, in my opinion, it is not clear where this conclusion comes from.
  6. Lines 223-227: the authors state that Eng expression is higher in the liver than in the lung and that this is maintained in vitro even after several passages. However, there is no quantitative data to support this claim except a picture of Eng expression in the lung and liver (unquantified). I believe that this claim should be demonstrated.
  7. Similarly, it is stated (line 248) that all cells have carried out gene recombination within the first 10h. However, this is not demonstrated by the data shown in Figure 5. To be able to affirm this, it would be necessary to show the analysis of the genomic DNA after 10h showing the disappearance of the floxed sequence.
  8. Figure 5E analyzes the effect of a complete or half reduction of Eng levels on signaling pathways. And this is relevant since part of the conclusions are based on these results. However, how has this half reduction been demonstrated? Treatment for 20h is used when in the only quantitative study of expression (Fig 5C) at 18h there is hardly any reduction in values and the decrease is seen at 24h. And these data only refer to lung, so we do not know what the dynamics will be in a tissue with higher basal levels such as liver. On the other hand, if it is based on the expression measured by WB, I think that quantification of Eng levels would be convenient to corroborate this.
  9. The authors state in the Discussion (line 312-314) that Eng heterozygosity is sufficient to develop vascular abnormalities in organs with low Eng levels but only in the presence of angiogenic stimuli, as opposed to the need for a second hit on the unmutated allele. Although in the general idea I agree I think that sentence is not fully demonstrated in the paper. For example, why do the heterozygous C57 mice have no phenotype and the 129/Ola do? Especially if there is no difference in expression (see point 3 of these comments). Or why in specific areas of certain patients? It is probably a somewhat more complex process than that sentence seems to convey, so I think perhaps it should be revisited.

Minor points

  1. Line 119: There is a typo: it should say GeNorm
  2. The statistical analyses in Figures 2B and 3A would have required the use of a two-way ANOVA.
  3. Table 1 does not show Eng primers.

Reviewer 3 Report

Summary

In this paper, the authors compare endoglin expression across several tissue types in mice, and hypothesize that lower basal endoglin underlies differences in VEGF signaling responses between different tissue beds.

Overall, the research question is interesting but the data presented lack important controls that would allow for meaningful interpretation of the data. Absent these, the data presented do not convincingly support authors’ conclusion (in Discussion) that a second hit that transiently or permanently eliminates the residual allele of endoglin in HHT is unnecessary to generate vascular malformations. Significant revision is necessary to address several major concerns in experimental approach and interpretation.

MAJOR CONCERNS

Differences in endoglin expression across tissue vascular beds is novel, and may contribute to the tissue specificity of vascular malformations in HHT1. However, there are several concerns in the interpretation of the data in this paper that undermine the overall study and its conclusions:

1) Authors perform RT-PCR on whole tissue digests from two mouse strains, and normalize endoglin amplification to that of putative endothelial markers. However, some of the markers chosen (e.g. ICAM, Tie2, etc) are not specific to endothelial cells and are also expressed in other cell types including in immune cells. Further, these reference markers themselves show tissue-specific differences. It is unclear why authors chose not to sort endothelial cells from different tissues, as this would have better controlled for input material.

2) Authors argue that low basal endoglin levels in skin, lung, intestine, and brain explain why these tissues are more prone to vascular malformation in HHT1. However, Figure 1D only shows significant differences between endoglin expression in other tissues compared to skin, and basal endoglin mRNA levels in lung, intestine and brain appear comparable to heart and kidney according to the presented data. This is similarly true for the data in Figure 2. 

3) In Figure 2, no EC marker is used to identify blood vessels in tissue sections, and so endoglin expression images cannot be compared to overall vascularity. This is especially relevant given that elsewhere authors say that endoglin expression is vessel type specific.

4) Images presented that are supposed to demonstrate differences in basal endoglin expression in human tissue from an HHT1 patient are unconvincing.

5) In Figure 5, authors isolate lung and liver EC from mouse, and assess endoglin expression and Akt signaling. However, these studies are done after 2-3 passages, and there is no indication that EC have retained their tissue-specific characteristics after this much time in culture. Authors show images that endoglin expression is lower in lung vs. liver EC in Figure 5B, but the same condition (0h, Figure 5D) shows a far less convincing set of images. Differences in Akt activation dynamics in Figure 5E are subtle, and it’s unclear how this would be able to so broadly explain the dramatic difference in vascular bed sensitivity to vascular malformation in HHT1.

MINOR CONCERNS

  • Figure 5’s immunofluorescent images are far too small and it is difficult to make out any of the details of the stains.
  • Line 228: “Taken advantage...” should read “Taking advantage...”
  • Line 288: “...were found exceptionally low...” should read “...were found to be exceptionally low...”

Round 2

Reviewer 2 Report

The article has improved a lot with the changes made and most of the questions have been answered. Congratulations!

Author Response

Thank you very much. We would like to thank the reviewer for his/her positive comment.

Reviewer 3 Report

We thank the reviewer for this comment and agree that alternative methods may also be used to quantify endoglin expression in multiple tissues including FACS sorting. However, tissue dissociation and endothelial cell isolation from multiple tissues are quite challenging. Specific protocols are required in order to reach sufficient mRNA yield for each organ. They include mechanical stress, enzymatic digestion at 37°C and additional intermediate purification steps that all could alter endoglin expression levels making the comparison between tissues difficult (Kalucka J. et al., 2020, Cell). Moreover, this approach requires the use of at least 6-8 mice per isolation making impossible the analysis of individual. As endoglin is principally expressed in vascular cells, we have therefore chosen to isolate mRNA from tissues and use GeNorm to identify the best reference pool for normalization. Although, we agree that the markers chosen are not strictly endothelial specific, they are widely used to label vascular cells and importantly by using GeNorm we have clearly validated that they were highly stable between all
conditions (stability M values) and that the reference pool chosen almost reached the threshold of 0.15 indicating that it could be used for normalization. Moreover, we have validated our results by analyzing tissue sessions stained for endoglin (Figure 2).

The logistical issues of sorting EC from multiple tissues pooled from multiple animals is well taken, and the use of GeNorm normalization as an alternative is acceptable given this context. Authors might consider including mention of this rationale in their main text given that the GeNorm approach is central to their study.

Nonetheless the point of concern remains: although the genes are suitable for normalization, inclusion of non-EC specific reference genes may result in normalization of Eng against a reference that includes non-EC contamination which might itself explain the apparent difference in Eng levels across tissues. This is of further concern given that in revision, authors clarify that two of the three selected reference genes are those with less EC specificity (Tie1, ICAM). Although authors show that M values are lowest for these reference genes indicating a lack of variability across tissues for these genes and that their mRNA data are validated at the protein level, their point would still be strengthened if they would consider instead using more classically EC-specific references, such as Cdh5 (VE-Cadherin), to show that normalization against more strictly EC-specific genes does not impact their results. In other words, authors should consider showing that selection of reference genes with greater EC-specific expression (e.g. Flk1, PECAM-1, Tie2, and/or Cdh5) replicates their results in Figure 1.  

This point is well taken and Figures 1 and 2 have been modified and include now statistics comparing the mean of each tissue with the mean of the other tissues using one-way ANOVA and Tukey’s multiple comparison tests confirming our findings that tissues with low basal levels of endoglin are more prone to develop vascular malformations in HHT1.

...The referee is correct although endoglin is a well-accepted vascular marker widely used to label blood vessels in tissue sections. However, since it is important that the difference in endoglin expression levels between the different tissues is absolutely clear to the general readers, we have selected another set of images showing Pecam1 and endoglin staining to illustrate this point. This is now included in the new Figure 2A and 2B. As expected, we show that Pecam1 expression levels do not vary while endothelial cells show striking heterogeneity in the levels of endoglin between tissues.

The inclusion of PECAM-1 staining as well as the improved statistics is much more convincing, and fully addresses my concerns.

Although, we had only limited material, we have now included a quantification of Pecam1 and
Endoglin expression levels in Figure 4A and 4B confirming our findings.

This fully addresses my concerns. Thank you.

We have now included a quantification of endoglin expression levels in Figure 5D and the gray values of the western blot of endoglin in Figure 5E to support our findings.

This is very helpful, thank you.

This is an important point. It was not our intention to claim that a fold increase of 1.5 will be enough to induce the formation of AVM in HHT1, although similar changes in VEGF signaling activity have already been reported in HHT mice developing AV shunts in retinas (Ola R et al., Nature Com 2016). Instead, our data indicate that in endothelial cells, endoglin haploinsufficiency leads to increased levels of akt activity only in specific vascular beds that might in turn be more sensitive to angiogenic triggers favoring the development of vascular malformations as we have previously demonstrated using HHT2 mice (Thalgott et al., Circulation, 2018). We have slightly modified the discussion as follow to make it clear for the readers.

“Our results also imply that mutations in endoglin leads to increased levels of Akt only in selected vascular beds that in turn might be more prone to respond inadequately to angiogenic triggers favoring the formation of vascular anomalies”.

This clarification is helpful, thank you.

TYPOGRAPHICAL ERRORS

An additional typographical error was detected upon re-review:

  • Line 105: "rational" should be "rationale"
